# Useless, or Untapped? Unlocking the Full Value of Zero-Advantage Samples for Better Policy Optimization

## Abstract

Reinforcement Learning with Verifiable Reward (RLVR) has emerged as a key technology to enhance the reasoning capabilities of large language models (LLMs). Recent studies have identified that the widespread prevalence of zero-advantage samples significantly impairs the training efficiency of RLVR algorithms, as the associated gradient vanishing prohibits effective parameter updates. To mitigate this issue, prior work attempts to discard such samples before or after rollout to improve efficiency. However, the computational cost incurred in generating these samples remains unavoidable. In this paper, we propose a novel perspective to address this challenge: if zero-advantage samples cannot be avoided, then we should leverage them. Specifically, we propose ZAPO, a **Z**ero-**A**dvantage sample-augmented **P**olicy **O**ptimization method that activates zero-advantage samples and enables them to make unique contributions to policy updates. Specifically, we utilize entropy to provide additional reward signals for zero-advantage samples, restoring their advantages, and thereby accelerating training efficiency. Simultaneously, entropy-based rewards drives exploration of previously unconsidered reasoning paths and expands the model's capability boundary. Experimental results on five math reasoning benchmarks and three base models (Qwen2.5-Math-1.5B, DeepSeek-R1-Distill-Qwen-1.5B, and Qwen2.5-Math-7B) demonstrate that ZAPO achieves superior average reasoning performance (45.7%, 54.2% and 55.4%), while achieving training acceleration factors of $1.7\times$, $1.3\times$ and $1.2\times$ in three base models, respectively, validating the effectiveness of the proposed approach.

## 1 Introduction

Test-time scaling has become a central research focus in the current large language model (LLM) community, aiming to guide the generation of appropriate chains of thought (CoT) to activate thinking process, and thereby enhance the reasoning capability of LLM. Recent advances introduce Reinforcement Learning from Verifiable Rewards (RLVR) as a flexible framework to realize this paradigm. By computing rule-based rewards for model-generated responses, reinforcement learning can directly and effectively fine-tune LLMs without requiring additional supervised data. The effectiveness of RLVR is guaranteed by policy optimization algorithms, such as Group Relative Policy Optimization (GRPO) (DeepSeek-AI et al., 2025). GRPO treats multiple sampled rollouts generated by the LLM for the same input as a group and computes relative rewards and advantages within each group. By maximizing token-level rewards, GRPO significantly improves the reasoning capacity of LLMs and has been widely adopted in advanced systems such as DeepSeek R1 (DeepSeek-AI et al., 2025) and Qwen3 (Yang et al., 2025).

Despite its considerable success, GRPO assigns zero advantage to rollout groups when faced with problems that are either fully within the model's mastery or entirely beyond its capacity. These samples, known as *zero-advantage samples*, contribute no gradients during training, thereby being regarded as ineffective samples. As training progresses, the proportion of zero-advantage samples increases substantially (as illustrated in Figure 2), significantly degrading GRPO's training efficiency. To address this limitation, recent approaches such as DAPO (Yu et al., 2025a) have proposed dynamic sampling strategies that oversample rollouts and filter out zero-advantage samples before advantage computation, preserving only effective samples for training. However, dynamic sampling requires

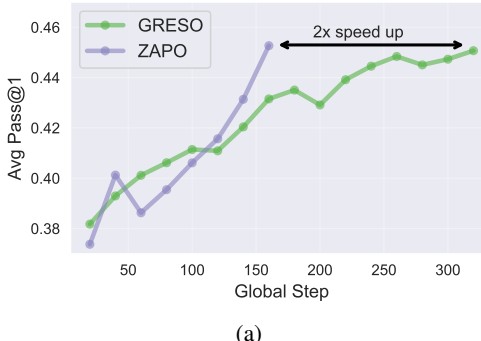 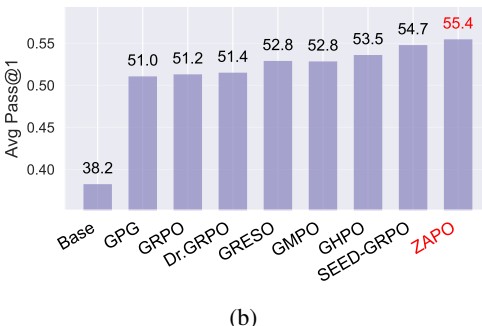

(a)             (b)

Figure 1: Comparison results of training efficiency and reasoning performance. (a) By activating zero-advantage samples, ZAPO achieves comparable performance to GRESO, which is based on GRPO, using only 50% of the training steps. (b) The utilization of zero-advantage samples enables ZAPO to attain optimal overall reasoning capabilities.

additional rollout generation to populate the training batch, introducing considerable computational overhead. To further enhance efficiency, GRESO (Zheng et al., 2025b) introduces pre-rollout filtering, which probabilistically filters out potentially over-difficult and over-simple prompts before rollout. The filtering probability is proportional to the estimated difficulty, derived from historical rewards. Nevertheless, as demonstrated in Figure 2, efficiency degradation persists despite pre-filtering, since zero-advantage samples are still probabilistically included, particularly in later runs.

Beyond the shortcomings of its policy optimization, RLVR also poses the risk of collapsing the capability boundary of the base model. Recent advances (Shao et al., 2024; Yue et al., 2025; Dong et al., 2025) suggest that RLVR tends to leverage existing reasoning paths within the base model rather than training the model to explore novel reasoning patterns. Additional evidence stems from the fact that the training of RLVR inherently leads to entropy collapse (Cui et al., 2025; Hao et al., 2025); when the entropy becomes excessively low, the model capacity for exploration is severely constrained (Yu et al., 2025a). To alleviate this issue, existing approaches (Dong et al., 2025; Liang et al., 2025; Liu et al., 2025c) have attempted to introduce extern data to increase the model's reasoning capability. These methods either construct additional datasets or integrate additional information into prompts. However, such auxiliary data increases training costs. The presence of zero-advantage samples suggests they can be treated as a rich pool of external data. Since they are not used in training, their latent reasoning patterns remain untapped by the model. Consequently, a natural question arises:

*Why not leverage these zero-advantage samples as auxiliary data to further enhance the model's reasoning capability?*

To this end, we propose Zero-Advantage sample-agmented Policy Optimization (ZAPO), a novel approach that incorporates zero-advantage samples into the training process to enhance training efficiency and expand capability boundaries. Specifically, we first divide the zero-advantage samples into two distinct classes: hard problems and easy problems. For easy problems, where the model has already mastered the fundamental reasoning patterns, we should encourage divergent thinking to explore a broader spectrum of potential solution. Conversely, for challenging problems that exceed the model's current capabilities, our primary objective is to guide the model in discovering one reasoning path that leads to correct solutions. Based on these insights, we propose Dual-level Adaptive Entropy Rewards (DAER) for zero-advantage samples during advantage computation, compelling them to generate effective gradients that facilitate policy model optimization. In addition, to strengthen the model's reasoning capabilities, we introduce Temporal Dynamic Advantage Reshaping (TDAR), which encourages the model to tackle more challenging problems within its competency range, thereby expanding its capability boundary. To comprehensively evaluate our method, we conducted experiments on five math reasoning benchmark datasets and three base models. The experimental results demonstrate that our method achieves superior comprehensive performance, exhibiting significant improvements on three models (3.2%, 4.2% and 13.0%) over GRPO, as shown in Figure 1 (b). Meanwhile, as illustrated in Figure 1 (a), our approach substantially improves training efficiency, achieving comparable performance to previous methods while requiring only 0.5× training steps.

Our contributions can be summarized as follows: 1) We propose a novel perspective for handling zero-advantage samples by leveraging them rather than circumventing them. 2) We introduce ZAPO, a zero-advantage sample-augmented policy optimization algorithm that activates zero-advantage samples by imposing additional adaptive entropy rewards. 3) Extensive experiments conducted on five datasets and three base models demonstrate ZAPO's improvements in both training efficiency and reasoning capabilities.

## 2 RELATED WORK

**RLVR for LLM Reasoning.** Reinforcement Learning with Verifiable Reward (RLVR), as an emerging post-training technique, has substantially improved LLM performance on complex reasoning tasks such as mathematics and programming. Unlike conventional RLHF (Ouyang et al., 2022), RLVR leverages simple, verifiable rule-based reward functions to compute rewards for model updating, thus eliminating dependence on supervised data. The success of RLVR has attracted considerable research interest, spawning a series of subsequent works that continuously refine and improve upon this paradigm. GRPO obviates the need for a value model by computing group relative advantages, thus enhancing training efficiency while maintaining effective training outcomes. Inspired by

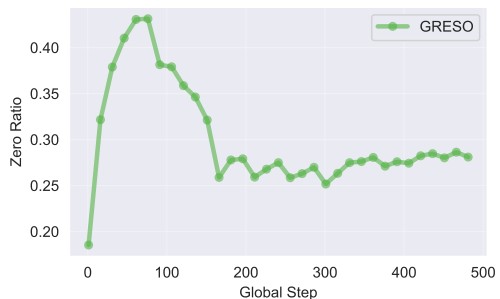

Figure 2: The ratio of zero-advantage samples in GRESO after applying the pre-filtering mechanism.

GRPO, some studies have proposed optimizations to policy optimization algorithms. For instance, to address GRPO's training instability on long-CoT Reasoning, DAPO (Yu et al., 2025a) introduces a token-level policy gradient loss that allocates differential attention to sequences of varying lengths, while GSPO Zheng et al. (2025a) mitigates the high variance inherent in GRPO's token-level accumulation through sequence-level importance sampling. Dr.GRPO (Liu et al., 2025b) propose an unbiased optimization approach that improves token efficiency while maintaining reasoning performance. There are some other works optimizing the training process of RLVR. For instance, AdapThink (Wan et al., 2025), LASER (Liu et al., 2025a), and VeriThinker (Chen et al., 2025b) design adaptive length-based reward that encourage models to generate tailored outputs of varying lengths for different problems, thereby improving both training efficiency and inference accuracy. In addition, RLPR (Yu et al., 2025b) and RLVMR (Zhang et al., 2025b) design problem-agnostic process rewards to improve generalizability for various tasks.

**Capability Boundary of LLMs in RLVR.** Despite the remarkable achievements, some studies (Havrilla et al., 2024; Team et al., 2025; Yue et al., 2025) have indicated that RLVR merely enables LLMs to rapidly identify pre-existing reasoning paths rather than fundamentally enhancing their reasoning capabilities. Consequently, when faced with problems that exceed their inherent capacity, RLVR does not facilitate the learning of novel problem-solving strategies. To address this limitation, RL-PLUS (Dong et al., 2025), SwS (Liang et al., 2025), and SRFT (Fu et al., 2025) incorporate extern datasets during training, enabling models to acquire new capabilities through reinforcement learning or supervised fine-tuning. However, constructing external datasets incurs substantial computational overhead and the development of high-quality datasets remains challenging. Alternatively, GHPO (Liu et al., 2025c) and MeRF (Zhang et al., 2025a) attempt to incorporate additional prompts to guide the models toward solving tasks beyond their current capabilities. Nevertheless, this approach introduces adverse effects on the generalizability of LLMs. Therefore, developing a simple yet effective method to enhance the capabilities of LLMs in RLVR remains an open research question.

**Zero-advantage Samples in RLVR.** Zero-advantage samples, defined as samples where the LLM's responses yield rewards of either all zeros or all ones, constitute a persistent challenge during RLVR training. Due to their zero advantages, such samples fail to produce effective gradients, thereby precluding parameter updates. Additionally, generating zero-advantage samples is computationally costly, and their incidence increases as training progresses, leading to substantial reductions in training

efficiency. A common approach is to discard these samples during training; for example, DAPO (Yu et al., 2025a) introduces dynamic sampling to exclude zero-advantage samples after generation, training only with effective samples. GRESO (Zheng et al., 2025b) and DEPO (Tang et al., 2025) further improve training efficiency by filtering out potentially trivial or excessively difficult problems prior to generation. Nevertheless, these methods do not eliminate the extra computational burden associated with generating zero-advantage samples. Furthermore, discarding these samples precludes the model from learning valuable reasoning strategies embedded within them, including fundamental knowledge from easy problems and advanced techniques from hard ones.

## 3 METHOD

### 3.1 PRELIMINARY

The Group Relative Policy Optimization (GRPO) is a variant of the Proximal Policy Optimization (PPO) algorithm (Yu et al., 2025a), proposed by DeepSeek (DeepSeek-AI et al., 2025) for fine-tuning the LLM to enhance its capability on reasoning tasks such as math and coding. Compared to PPO, the core advantage of GRPO lies in utilizing the mean reward of a group of rollouts as a baseline to estimate the relative rewards and advantages of each sample within the group, thereby eliminating the value network and significantly improving training efficiency and stability. Specifically, for an input prompt $x$, GRPO first employs the policy model $\pi_{old}$ to sample $G$ responses as a rollout group $\{o_1, o_2, ..., o_G\}$ and computes the corresponding rewards $\{r_1, r_2, ..., r_G\}$ using rule-based reward functions. GRPO then optimizes the policy model $\pi_\theta$ by maximizing the following objective:

$$\mathcal{J}_{\text{GRPO}}(\theta) = \mathbb{E}_{x \sim \mathcal{D}, \{o_i\}_{i=1}^G \sim \pi_{\theta_{\text{odd}}}(\cdot|x)}$$

$$\left[ \frac{1}{G} \sum_{i=1}^G \frac{1}{|o_i|} \sum_{t=1}^{|o_i|} \min\left( w_{i,t}(\theta)\widehat{A}_{i,t}, \ \text{clip}\left(w_{i,t}(\theta), 1-\varepsilon, 1+\varepsilon\right)\widehat{A}_{i,t} \right) \right], \tag{1}$$

where the importance ratio $w_{i,t}$ and the advantage $\widehat{A}_{i,t}$ are defined as:

$$w_{i,t}(\theta) = \frac{\pi_\theta(y_{i,t}|x, y_{i,<t})}{\pi_{\theta_{\text{old}}}(y_{i,t}|x, y_{i,<t})}, \quad \widehat{A}_{i,t} = \frac{r_i - \text{mean}\left(\{r_i\}_{i=1}^G\right)}{\text{std}\left(\{r_i\}_{i=1}^G\right)}. \tag{2}$$

Although group-normalized relative rewards effectively assess rollout advantages, identical rewards within a group result in every rollout being assigned zero advantage. Based on Equation (1), the gradient update for GRPO can be formalized as follows:

$$\nabla_\theta \mathcal{J}_{\text{GRPO}}(\theta) = \nabla_\theta \mathbb{E}_{x \sim \mathcal{D}, \{o_i\}_{i=1}^G \sim \pi_{\theta_{\text{old}}}(\cdot|x)} \left[ \frac{1}{G} \sum_{i=1}^G \frac{1}{|o_i|} \sum_{t=1}^{|o_i|} w_{i,t}(\theta)\widehat{A}_{i,t} \right]. \tag{3}$$

As illustrated in Equation (3), when $\widehat{A}_{i,t} = 0$, the corresponding GRPO gradient vanishes, rendering the rollouts in this group ineffective for updating the policy model $\pi_\theta$. Some recent methods (Yu et al., 2025a; Zheng et al., 2025b; Tang et al., 2025) design data selection strategies that filter effective samples for training to improve training efficiency. Nevertheless, due to the dynamic evolution of model capabilities, zero-advantage samples remains unavoidable. As illustrated in Figure 2, despite implementing pre-filtering mechanisms, GRESO (Zheng et al., 2025b) still samples a substantial number of zero-advantage samples during the rollout process, impeding further efficiency improvements. In this paper, we attempt to transform these zero-advantage samples into effective samples to facilitate model training.

### 3.2 DUAL-LEVEL ADAPTIVE ENTROPY REWARD FOR ZERO-ADVANTAGE SAMPLES

Given that the direct cause of gradient vanishing stems from all samples receiving identical rewards, an intuitive approach involves assigning additional rewards to these samples to ensure that they possess non-zero advantages. However, hard and easy problems exhibit substantial differences, and directly allocating rewards to both categories may lead to training collapse. This raises a critical question: how can we appropriately distribute rewards among different zero-advantage samples?

In this work, we design a dual-level adaptive entropy reward mechanism to achieve this objective, allocating rewards to zero-advantage samples from both prompt-level and sequence-level perspectives. Inspired by prior work (Kuhn et al., 2023; Farquhar et al., 2024; Chen et al., 2025a), we leverage the semantic entropy of responses generated by LLMs for each prompt to compute prompt-level entropy rewards. Semantic entropy (SE) (Kossen et al., 2024; Farquhar et al., 2024) is an entropy-based metric for quantifying semantic diversity of responses within one group. A low semantic entropy suggests that the LLM may be constrained to a single fixed reasoning path, whereas excessively high entropy typically indicates that the given prompt extends beyond the LLM's capacity. Consequently, we implement differentiated treatments for hard and easy problems: we encourage entropy increase in responses to easy questions to cultivate divergent thinking capabilities, while constraining entropy to complex problems to facilitate more lucid reasoning and accurate solutions for hard queries.

Specifically, we begin by computing the semantic entropy of the sampled prompts. The definition of semantic entropy (Farquhar et al., 2024) can be formalized as:

$$\text{SE}(x) = -\sum_c P(c|\boldsymbol{x})\log P(c|\boldsymbol{x}) = -\sum_c \left(\left[\sum_{o_i \in c} P(o_i|x)\right] \log \left[\sum_{o_i \in c} P(o_i|x)\right]\right), \quad (4)$$

where $P(o_i|x)$ denotes the probability of generating response $o_i$ for prompt $x$ under the policy model $\pi_{\theta_{old}}$, and $c$ represents a cluster of semantically similar responses.

Theoretically, we should enumerate the entire space of potential responses to determine $P(o_i|x)$ and all possible semantic clusters; however, it's computationally intractable. Consequently, following prior work (Kossen et al., 2024; Farquhar et al., 2024), we estimate the semantic entropy in Equation (4) using a Monte Carlo integration. For a given group of responses $\{o_1, o_2, ..., o_G\}$, we first employ sequence embeddings to cluster them into distinct semantic clusters. The sequence embedding for each response is obtained via mean pooling of the hidden embeddings from the final layer of LLM:

$$\boldsymbol{h}_{o_i} = \frac{1}{|o_i|} \sum_{t=1}^{|o_i|} \boldsymbol{h}_{i,t}, \quad (5)$$

where $\boldsymbol{h}_{i,t}$ represents the hidden embedding of the $t$-th token in response $o_i$. Subsequently, we apply the K-Means algorithm to cluster the sequence embeddings $\{\boldsymbol{h}_{o_i}\}_{i=1}^{G}$ into distinct clusters. Then, we estimate the semantic entropy using the following formula:

$$\text{SE}(x) \approx -\sum_{i=1}^{|C|} P(C_i|x)\log P(C_i|x), \quad (6)$$

where $C_i$ denotes the $i$-th cluster and $P(C_i|x) = \sum_{o_j \in C_i} \pi_{\theta_{old}}(o_i|x)$ denotes the probability of generating responses within the cluster $C_i$.

Semantic entropy evaluates the diversity of an LLM's responses to a particular question from the prompt-level perspective; however, it lacks the capacity for fine-grained assessment at the level of individual responses. To address this limitation, we introduce sequence-level entropy rewards based on the token-level entropy of each response. Specifically, for a response $o_i = \{y_{i,1}, y_{i,2}, ..., y_{i,|o_i|}\}$, we compute its sequence-level entropy as follows:

$$\text{TE}(o_i) = -\frac{1}{|o_i|} \sum_{t=1}^{|o_i|} \sum_{y \in \mathcal{V}} \pi_{\theta_{old}}(y|x, y_{i,<t})\log \pi_{\theta_{old}}(y|x, y_{i,<t}), \quad (7)$$

where $\mathcal{V}$ denotes the vocabulary set. Finally, the entropy of each response for prompt $x$ is calculated by jointly considering both semantic entropy and token-level entropy as: $E_i = (SE(x) + TE(o_i))/2$.

Based on the derived entropy, we compute the reward for each zero-advantage sample as follows:

$$r'_i = \log(1 + E_i) \cdot \mathbb{I}_i. \quad (8)$$

where $\mathbb{I}_i$ is an indicator function defined as: $\mathbb{I}_i = \begin{cases} 1, & \text{if } r_i = 1 \\ -1, & \text{if } r_i = 0.1 \end{cases}$. In this paper, we follow previous work (DeepSeek-AI et al., 2025; Yu et al., 2025a; Zheng et al., 2025b) by setting the reward

for correct responses to 1 and the reward for incorrect responses to 0.1. Therefore, $\mathbb{I}_i$ indicates whether the sample corresponds to a hard or easy problem. For a easy zero-advantage sample, the definition of entropy ensures that $E_i$ is a non-negative value, resulting in non-negative entropy rewards. According to Equation (8), easy samples with higher entropy receive greater entropy rewards, encouraging the model to explore more diverse responses. For hard samples, previous work (Zhu et al., 2025) has observed the remarkable effectiveness of negative sample reinforcement learning.Building upon this insight, we assign non-positive rewards to hard samples, thereby compelling the model to seek new reasoning paths to address these challenging questions. When a hard sample exhibits higher entropy, according to Equation (8), it receives a larger negative reward, as elevated entropy indicates that the LLM is likely producing uncertain or incoherent outputs, which should be preferentially discarded.

During the early stages of training, due to the limited capability of the model, both hard and easy problems may exhibit high entropy, leading to excessively large entropy rewards for zero-advantage samples and consequently causing training instability. Additionally, we wish to encourage the model to focus more on hard questions in order to improve its reasoning capability. Based on these considerations, we reformulate Equation (8) as an adaptive entropy reward:

$$\hat{r}_i = r'_i \cdot \frac{t}{T} \cdot \delta \tag{9}$$

where $t$ denotes the current training step, $T$ represents the total training steps. $\delta$ is a parameter used to control the update weights between hard and easy samples, and is computed as follows:

$$\delta = \begin{cases} p_{easy}/(p_{easy} + \alpha \cdot p_{hard}), & \text{if } r_i = 1 \\ 1 - p_{easy}/(p_{easy} + \alpha \cdot p_{hard}), & \text{if } r_i = 0.1 \end{cases}. \tag{10}$$

where $p_{easy}$ and $p_{hard}$ denote the proportions of easy and hard zero-advantage samples in the training batch, respectively. $\alpha$ is a hyperparameter that controls the relative weight of hard samples. By incorporating the factor $\frac{t}{T}$, we ensure that the entropy rewards for zero-advantage samples gradually increases as training progresses, thereby enhancing training stability. Furthermore, by adjusting the value of $\alpha$, we can modulate the relative importance of hard samples, thereby encouraging the model to prioritize learning from challenging problems.

### 3.3 TEMPORALLY DYNAMIC ADVANTAGE RESHAPING

The activation of zero-advantage samples enables the model to explore correct reasoning paths for previously unsolved problems. Once the correct solution for a complex problem is identified, we expect the model to rapidly master it to enhance its reasoning capabilities. Some recent work (Zhu et al., 2025; Liu et al., 2025c; Zhang & Zuo, 2025; Liang et al., 2025) has demonstrated that training with more challenging samples can effectively extend the capability boundary of LLM. Motivated by the above considerations and prior research, we introduce temporal dynamic advantage reshaping, which progressively intensifies the focus on hard samples as training advances. Specifically, we first compute the difficulty coefficient $d(x)$ for each sample with non-zero advantage as follows:

$$d(x) = 1 - \frac{1}{G} \sum_{i=1}^{G} r_i. \tag{11}$$

Then we reshape the advantage of each sample as follows:

$$\widetilde{A}_{i,t} = \widehat{A}_{i,t} \cdot (1 + \frac{1}{1 + e^{-\beta \cdot \frac{t}{T} \cdot (d(x) - 0.5)}}), \tag{12}$$

where $\beta$ is a hyperparameter that controls the preference for difficult samples. By incorporating the temporal factor $\frac{t}{T}$, we ensure that the preference for difficult samples gradually increases as training progresses, thereby enhancing training stability. Based on Equation (12), the update gradients for samples with non-zero advantages can be computed as:

$$\nabla_\theta \mathcal{J}_{\text{ZAPO}}(\theta) = \nabla_\theta \mathbb{E}_{x \sim \mathcal{D}, \{o_i\}_{i=1}^{G} \sim \pi_{\theta_{\text{old}}}(\cdot|x)}$$

$$\frac{1}{G} \sum_{i=1}^{G} \frac{1}{|o_i|} \sum_{t=1}^{|o_i|} w_{i,t}(\theta) \underbrace{\left[ \widehat{A}_{i,t} \cdot (1 + \frac{1}{1 + e^{-\beta \cdot \frac{t}{T} \cdot (d(x) - 0.5)}}) \right]}_{\widetilde{A}_{i,t}}. \tag{13}$$

In the initial phase of training, the introduction of $\frac{t}{T}$ ensures that hard and easy samples receive Approximately equal gradient updates, allowing the model to acquire adequate fundamental capabilities. As training proceeds, the gradient updates for hard samples are gradually strengthened, which helps the model achieve superior reasoning performance.

Finally, we compute the advantages of all samples as: $\bar{A}_{i,t} = \begin{cases} \widetilde{A}_{i,t}, & \text{if } \widehat{A}_{i,t} \neq 0 \\ \hat{r}_i, & \text{if } \widehat{A}_{i,t} = 0 \end{cases}$.

To further enhance training stability, we introduce the pre-filtering mechanism proposed by GRESO, which dynamically adjusts the sampling proportions of hard and easy problems through the assignment of adaptive sampling probabilities. The detailed sampling procedure and algorithm can be found in GRESO (Zheng et al., 2025b).

## 4 EXPERIMENTS

### 4.1 EXPERIMENTAL SETUP

**Models &Datasets.**   Following the same settings as GRESO (Zheng et al., 2025b), We conduct experiments on three widely used base models: Qwen2.5-Math-1.5B (Yang et al., 2024), DeepSeek-R1-Distill-Qwen-1.5B (DeepSeek-AI et al., 2025), and Qwen2.5-Math-7B (Yang et al., 2024). We train the aforementioned base models on the DAPO Math (Yu et al., 2025a) and Light-eval (Hendrycks et al., 2021) datasets, consistent with GRESO. To evaluate the performance of trained models on complex mathematical reasoning tasks, we select five mathematical reasoning benchmark datasets, including Math500 (Lightman et al., 2024), AIME24, AMC, Minerva Math (Lewkowycz et al., 2022) and Olympiad Bench (Huang et al., 2024).

**Training & Evaluation Details.**   We implement our method under the verl Sheng et al. (2025) framework. We set the maximum training steps to 1000, perform evaluation on the five datasets every 20 steps. We set $\alpha$ to 2 for Qwen2.5-Math-1.5B, and 5 for DeepSeek-R1-Distill-Qwen-1.5B and Qwen2.5-Math-7B. $\beta$ is set to 5 for all models. We employ the AdamW optimizer (Loshchilov & Hutter, 2019) with a learning rate of 1e-6 and a weight decay of 0.01. We train Qwen2.5-Math-1.5B on 4 A100 GPUs and Qwen2.5-Math-7B and DeepSeek-R1-Distill-Qwen-1.5B on 8 H800 GPUs. Similar to GRESO, we set the temperature to 1 for all models and use pass@1 as the assessment metric for evaluation. Each evaluation for all benchmarks is repeated 4 times to ensure the stability and reliability of the results. More training and evaluation details can be found in the appendix B.

**Baselines.**   To demonstrate the performance advantage, we select 7 recent reinforcement learning methods as baselines, including GRPO (Shao et al., 2024), Dr.GRPO (Liu et al., 2025b), GPG (Chu et al., 2025), SEED-GRPO (Chen et al., 2025a), GRESO (Zheng et al., 2025b), GHPO (Liu et al., 2025c) and GMPO (Zhao et al., 2025). In addition, we conduct a comprehensive comparison with the closely related baseline, GRESO (Zheng et al., 2025b), to highlight that ZAPO achieves improvements not only on performance but also on training efficiency.

### 4.2 MAIN RESULTS

**Overall Performance.**   We present the comparative experimental results of ZAPO against other baselines across multiple mathematical reasoning datasets in Table 1. The experimental results for other methods are sourced from SEED-GRPO (Chen et al., 2025a) or their original papers (Liu et al., 2025b;c), except for GRESO (Zheng et al., 2025b). We train GRESO under identical settings to ensure a more fair comparison. As shown in Table 1, our method achieves optimal overall performance across various base models and reasoning datasets, demonstrating significant improvements in reasoning performance. Compared to the base models, we achieve average performance improvements of 18.8 %, 15.2 %, and 12.8 % across the three base models, respectively.

Compared to GRESO, our method demonstrates superior reasoning performance across different base models. Notably, on more challenging datasets such as AIME, our method achieves substantial improvements over GRESO, with gains of 5.0%, 9.8%, and 6.6 % on the three models. This indicates that utilizing the zero-advantage samples, particularly hard samples that GRESO discards, can effectively help the model solve complex problems and thereby expand its capability boundary.

Table 1: Comparison of Pass@1 performance across five mathematical reasoning benchmarks.

| Method | AMC | Math500 | Miner. | Olymp. | AIME24 | Avg. |
|---|---|---|---|---|---|---|
| *Qwen2.5-Math-base-7B* | | | | | | |
| Base Model (Yang et al., 2024) | 38.5 | 53.3 | 17.8 | 29.9 | 0.0 | 27.9 |
| GRPO-1.5B (Shao et al., 2024) | 49.4 | 75.2 | 25.7 | 39.0 | 10.0 | 42.5 |
| Dr.GRPO-1.5B (Liu et al., 2025b) | 53.0 | 74.2 | 25.7 | 37.6 | 20.0 | 42.1 |
| SEED-GRPO-1.5B (Chen et al., 2025a) | 50.6 | 75.4 | 26.8 | 41.3 | 23.3 | 43.5 |
| GRESO-1.5B (Zheng et al., 2025b) | 61.4 | 76.6 | 33.3 | 38.5 | 15.0 | 45.0 |
| GMPO-1.5B (Zhao et al., 2025) | 53.0 | 77.6 | 30.1 | 38.7 | 20.0 | 43.9 |
| ZAPO-1.5B | 61.1 | 77.4 | 29.8 | 40.1 | 20.0 | **45.7** |
| *Qwen2.5-Math-base-7B* | | | | | | |
| Base Model (Yang et al., 2024) | 44.3 | 74.0 | 21.7 | 39.5 | 16.7 | 39.2 |
| GRPO-7B (Shao et al., 2024) | 59.0 | 83.4 | 32.4 | 41.3 | 40.0 | 51.2 |
| Dr.GRPO-7B (Liu et al., 2025b) | 62.7 | 80.0 | 30.1 | 41.0 | 43.3 | 51.4 |
| GPG-7B (Chu et al., 2025) | 65.0 | 80.0 | 34.2 | 42.4 | 33.3 | 51.0 |
| GHPO-7B (Liu et al., 2025c) | 70.0 | 82.2 | 38.2 | 45.3 | 31.9 | 53.5 |
| SEED-GRPO-7B (Chen et al., 2025a) | 64.7 | 82.2 | 35.0 | 45.2 | 43.3 | 54.7 |
| GRESO-7B (Zheng et al., 2025b) | 68.1 | 81.6 | 34.7 | 43.9 | 35.8 | 52.8 |
| GMPO-7B (Zhao et al., 2025) | 61.4 | 82.0 | 33.5 | 43.6 | 43.3 | 52.7 |
| ZAPO-7B | 70.8 | 80.7 | 34.8 | 44.0 | 46.6 | **55.4** |
| *Deepseek-R1-Distill-Qwen-1.5B* | | | | | | |
| Base Model (Zheng et al., 2025b) | 50.3 | 75.4 | 26.5 | 37.3 | 16.7 | 41.2 |
| GRESO-1.5B (R1-Distill) (Zheng et al., 2025b) | 63.8 | 84.3 | 32.1 | 49.2 | 30.0 | 51.9 |
| ZAPO-1.5B (R1-Distill) | 66.3 | 84.1 | 36.4 | 47.7 | 36.6 | **54.2** |

Compared to GHPO-7B, which incorporate reference hints for hard problems during training, our ZAPO-7B achieves a substantial performance improvement of up to 14.7% on AIME. This suggests that directing greater attention to challenging problems and allowing the model to learn them independently may be a more effective approach for expanding the capability boundary than directly having the model memorize reference solutions to difficult problems.

Table 2: Comparison with GRESO on training efficiency. Seconds for Rollout and Training time, hours for others.

| Method | Rollout | Training | Steps | Total Rollout | Total Training | Total Time | Avg. |
|---|---|---|---|---|---|---|---|
| *Qwen2.5-Math-1.5B* | | | | | | | |
| GRESO | 100.4 | 104.3 | 320 | 8.9 | 9.2 | 18.1 | 45.4 |
| ZAPO | 135.5 | 112.8 | 160 | 6 | **5.0 (1.8×)** | **11.0 (1.7×)** | 45.3 |
| *Qwen2.5-Math-7B* | | | | | | | |
| GRESO | 76.5 | 84.2 | 440 | 9.4 | 10.2 | 19.6 | 55.6 |
| ZAPO | 89.6 | 96.1 | 320 | 8.0 | **8.5 (1.2×)** | **16.5 (1.2×)** | 56.8 |
| *Deepseek-R1-Distill-Qwen-1.5B* | | | | | | | |
| GRESO | 144.4 | 76.1 | 280 | 11.2 | 6.0 | 17.0 | 55.9 |
| ZAPO | 187.8 | 97.0 | 160 | 8.3 | **4.3 (1.4×)** | **12.6 (1.3×)** | 56.4 |

**Training Efficiency.** We compare the training efficiency of ZAPO with GRESO in Table 2 and Figure 1 (a). As shown in Table 2, ZAPO significantly reduces the total training time compared to GRESO across different base models, achieving speedups of 1.7×, 1.2×, and 1.3×, respectively. GRESO improves training efficiency by filtering zero-advantage samples to increase the utilization rate of effective samples. However, as training progresses, zero-advantage samples inevitably become part of the training data, resulting in the model learning only a subset of samples at each training step, as illustrated in Figure 2. In contrast, ZAPO assigns additional rewards to zero-advantage samples,

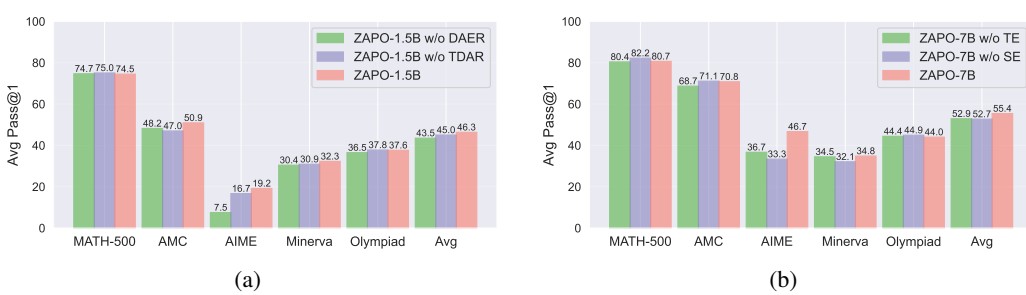

Figure 3: Ablation study results across five datasets on Pass@1 are presented as follows: (a) Ablation of DDAR and TDAR on Qwen2.5-Math-1.5B; (b) Ablation of semantic entropy and token-level entropy on Qwen2.5-Math-7B.

enabling all samples to be fully leveraged at each training step. As a result, ZAPO can achieve comparable or even better reasoning performance with fewer training steps than GRESO.

### 4.3 ABLATION STUDY

**Ablation of DAER and TDAR.** To validate the effectiveness of the two key components proposed in this work, dual-level adaptive entropy reward and temporal dynamic advantage reshaping, we construct two variants, w/o DAER and w/o DDAR, and train Qwen2.5-Math-1.5B for 400 steps under identical settings. The experimental results are shown in Figure 3 (a). After removing the respective components, both w/o DAER and w/o DDAR exhibit significant performance drops, confirming the effectiveness of our method. Furthermore, on challenging datasets, particularly AIME, w/o DAER and w/o DDAR show significant decreases of 11.7 % and 2.5 %, respectively. These findings further demonstrate that leveraging zero-advantage samples and increasing attention to difficult problems effectively expand the model's capability boundary.

**Ablation of Semantic and Token-level Entropy.** To validate the effectiveness of the dual-layer entropy reward mechanism, we construct two variants: ZAPO w/o TE, which utilizes only semantic entropy for reward computation, and ZAPO w/o SE, which employs only token-level entropy. The experimental results on Qwen2.5-Math-7B are presented in Figure 3 (b). Both ZAPO w/o TE and ZAPO w/o SE exhibit performance drops compared to ZAPO, confirming the effectiveness of the dual-level entropy reward. Notably, ZAPO w/o SE demonstrates a more pronounced performance decline, particularly on challenging datasets such as AIME, where it experiences a significant drop of 13.4%. This suggests that semantic entropy, which captures the diversity of responses at the prompt level, plays a crucial role in effectively activating hard zero-advantage samples.

Additional detailed analytical experiments can be found in Appendix C and Appendix D.

## 5 CONCLUSION

In this work, we propose ZAPO, a Zero-Advantage sample-enhanced Policy Optimization method to address the prevalent and unavoidable zero-advantage samples in GRPO training. Since zero-advantage samples do not produce effective gradients, they pose a significant obstacle to training efficiency. By assigning dual-layer adaptive entropy rewards to zero-advantage samples, ZAPO effectively activates these samples and leverages them to enhance both training efficiency and reasoning performance. To further accelerate training efficiency and extend the capability boundary, we introduce a temporally dynamic advantage reshaping mechanism that adaptively guides the model to increase its focus on challenging problems as training progresses. Experimental results across multiple base models and mathematical reasoning datasets demonstrate that our method achieves superior comprehensive reasoning performance, with particularly notable improvements on difficult datasets such as AIME. Additionally, our method achieves training speedups of $1.7\times$, $1.2\times$, and $1.3\times$ on three various base models, respectively, validating the effectiveness of the proposed ZAPO.

ETHICS STATEMENT

This research adheres to strict ethical standards throughout the study. All experiments are conducted on publicly available datasets (DAPO Math and Light-eval) and pre-trained models (Qwen2.5-Math-1.5B, DeepSeek-R1-Distill-Qwen-1.5B, and Qwen2.5-Math-7B), ensuring no privacy concerns or sensitive information is involved. The study does not generate or utilize any fabricated data, and all reported results are reproducible. We have properly cited and acknowledged all relevant prior work and baseline methods used in our comparisons. The research methodology focuses solely on algorithm optimization for reinforcement learning with verifiable rewards in mathematical reasoning tasks, without involving human subjects, animal experiments, or any activities requiring ethical review. Our work aims to improve computational efficiency and reasoning capabilities of large language models, contributing positively to the advancement of artificial intelligence research. No conflicts of interest exist that could compromise the integrity of this research.

REPRODUCIBILITY STATEMENT

Details of our experimental setup are provided in Section 4.1 and Appendix B. All resources utilized in this work, including datasets and base models including Qwen2.5-Math-1.5B, DeepSeek-R1-Distill-Qwen-1.5B, and Qwen2.5-Math-7B, are publicly accessible. Our implementation code will be made publicly available on GitHub upon acceptance of the paper.

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

## A    USE OF LLMS

In this work, we employ LLMs only for refining the completed manuscript, aiming to reduce grammatical errors and enhance coherence. No original content is generated by the models in any part of this study.

## B    EXPERIMENTAL SETUP

**Models & Datasets.**    Following the same settings as GRESO (Zheng et al., 2025b), We conduct experiments on three widely used base models: Qwen2.5-Math-1.5B (Yang et al., 2024), DeepSeek-R1-Distill-Qwen-1.5B (DeepSeek-AI et al., 2025), and Qwen2.5-Math-7B (Yang et al., 2024). We train the aforementioned base models on the DAPO Math (Yu et al., 2025a) and Light-eval (Hendrycks et al., 2021) datasets, consistent with GRESO (Zheng et al., 2025b). DAPO Math is a mathematical reasoning dataset proposed by Yu et al. (2025a), comprising 17.9k samples with integer solutions, while Light-eval consists of 7.5k problems with LaTeX-formatted solutions. To evaluate the performance of trained models on complex mathematical reasoning tasks, we select five mathematical reasoning benchmark datasets: 1) Math500 (Lightman et al., 2024), a random selected subset of 500 problems from Light-eval. 2) AIME24, contains 30 high-school level olympiad problems from the American Invitational Mathematics Examination 2024. 3) AMC, consists of 83 multiple-choice problems with intermediate difficulty from the AMC series. 4) Minerva Math (Lewkowycz et al., 2022), a collection of 272 graduate-level problems requiring multi-step reasoning. 5) Gaokao (Zhang et al., 2023), includes 2,811 questions from GAOKAO exams between 2010 and 2022. 6) Olympiad Bench (Huang et al., 2024), contains 675 high-difficulty olympiad problems.

**Training & Evaluation Details.** We implement our method under the verl Sheng et al. (2025) framework. Following GRESO's setup, we utilize vllm (Kwon et al., 2023) for rollout. During training, we set the context length to 4096 for Qwen2.5-Math-7B and Qwen2.5-Math-1.5B, with maximum prompt length and maximum response length of 1536 and 2560, respectively. For DeepSeek-R1-Distill-Qwen-1.5B, we set the context length to 8192, with prompt length and response length of 2048 and 6144, respectively. We set the maximum training steps to 1000, perform evaluation on the five datasets every 20 steps, and retain the checkpoint with the highest average score. The training batch size is set to 256, with each prompt sampling 8 responses. We set $\alpha$ to 2 for Qwen2.5-Math-1.5B, and 5 for DeepSeek-R1-Distill-Qwen-1.5B and Qwen2.5-Math-7B. The hyperparameters $\beta$ is set to 5 for all models. We employ the AdamW optimizer (Loshchilov & Hutter, 2019) with a learning rate of 1e-6 and a weight decay of 0.01. We train Qwen2.5-Math-1.5B on 4 A100 GPUs and Qwen2.5-Math-7B and DeepSeek-R1-Distill-Qwen-1.5B on 8 H800 GPUs. Similar to GRESO (Zheng et al., 2025b), we set the temperature to 1 for all models and use pass@1 as the assessment metric for evaluation. Each evaluation for all benchmarks is repeated 4 times to ensure the stability and reliability of the results. More training and evaluation details can be found in the appendix.

**Baselines.** To demonstrate the performance advantage, we select 6 recent reinforcement learning methods as baselines, including GRPO (Shao et al., 2024), Dr.GRPO (Liu et al., 2025b), GPG (Chu et al., 2025), SEED-GRPO (Chen et al., 2025a), GRESO (Zheng et al., 2025b), GHPO (Liu et al., 2025c) and GMPO (Zhao et al., 2025). In addition, we conducted a comprehensive comparison with the closely related baseline, GRESO (Zheng et al., 2025b), to highlight that ZAPO achieves improvements not only in performance, but also in training efficiency.

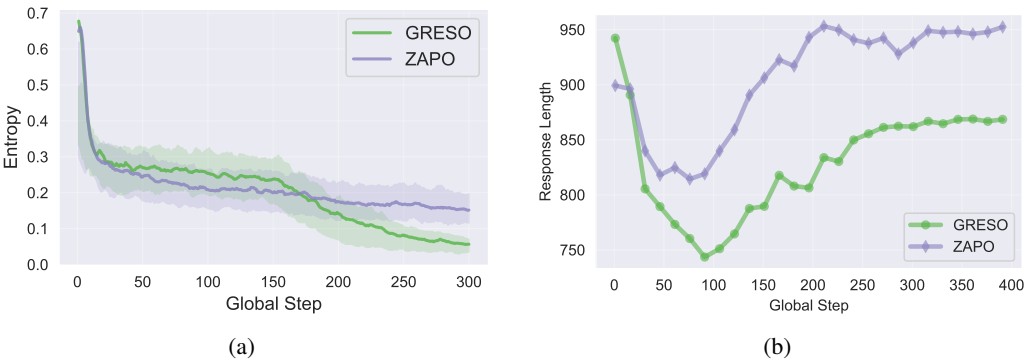

(a)  (b)

Figure 4: Analysis of entropy and response length. (a) Comparison of token-level entropy between ZAPO and GRESO, where the line represents the average entropy across all samples, and the upper and lower bounds of the shaded region represent the entropy of difficult and simple problems, respectively; (b) Comparison of response length between ZAPO and GRESO.

## C  ANALYSIS OF ENTROPY.

Figure 4 (a) illustrates the comparison of token-level entropy between ZAPO and GRESO during the training of Qwen2.5-Math-1.5B. The results demonstrate that ZAPO maintains higher entropy throughout the training process, indicating that encouraging entropy increase on simple problems effectively mitigates the entropy collapse issue. Additionally, it can be observed that ZAPO demonstrates lower entropy compared to GRESO between steps 30 and 180. This is attributable to the dual-layer adaptive entropy reward, which suppresses the entropy associated with a large number of challenging examples encountered in the early stages of training. As training progresses, an increasing proportion of simple problems is sampled. By encouraging entropy increase on these examples, ZAPO not only mitigates entropy collapse but also effectively preserves the model's exploratory potential, which may serve as a crucial factor in expanding the model's capability boundary.

## D ANALYSIS OF RESPONSE LENGTH.

Figure 4 (b) presents a comparison of response lengths between ZAPO and GRPO during the training of Qwen2.5-Math-7B. The results indicate that ZAPO generates longer average response lengths during training, which stems from ZAPO's encouragement of enhanced exploration capabilities and increased attention to difficult problems. The extended response length enables the model to engage in more comprehensive reasoning when encountering challenging problems, thereby increasing the likelihood of identifying correct solutions and consequently improving the model's reasoning capabilities.

