# OpenReview forum: "Useless, or Untapped? Unlocking the Full Value of Zero-Advantage Samples for Better Policy Optimization"
_ICLR.cc/2026/Conference — ICLR 2026 Conference Withdrawn Submission_

### Official Review · Reviewer_wyWp · 2025-10-26

**Soundness:** 1
**Presentation:** 2
**Contribution:** 2
**Rating:** 2
**Confidence:** 4

**Summary:**

This work augments GRPO-style training with a semantic entropy signal to break ties in zero-advantage groups. For each prompt, it clusters $G$ sampled rollouts in embedding space and computes entropy over cluster masses to reward diverse solution modes. It also introduces a batch-adaptive reweighing using proportions of all-correct versus all-incorrect zero-advantage groups to shift learning toward harder problems. The method is lightweight and plug-compatible, requiring no additional labels and minimal code changes. Empirically, it aims to promote exploration and stabilize training signals when rewards alone provide no gradient.

**Strengths:**

- Lightweight semantic entropy signal to differentiate otherwise identical-reward rollouts, helping the update focus on genuinely diverse solution modes without extra labels or major code changes.
- Overall topic is timely and important.
- Relatively clear writing.

**Weaknesses:**

- <1> Unverified claim:

(1) You claim that continuing to train on all-solved groups *“encourages divergent thinking to explore a broader spectrum of potential solutions.”* Please provide empirical evidence that this yields more diverse responses. I suggest an ablation that trains only on all-correct-group samples using your proposed entropy score, and then measures whether policy entropy increases. Additionally, report diversity metrics (e.g., policy entropy) to substantiate the claim.

(2) Arbitrary design choices without supporting ablations. Several hyperparameters and schedules appear ad hoc, with no ablation to justify them. For example, how should $\alpha$ be chosen to *“encourage the model to prioritize learning from challenging problems”*? Why is it set to $2$ and $5$ for different models, and how should it be tuned in practice? Moreover, what happens if the $\tfrac{t}{T}$ factor is removed? These are just a few examples. The lack of systematic ablations makes it unclear which components actually help. I recommend following the style of Table 1 in DAPO (https://arxiv.org/pdf/2503.14476) to present clear, controlled ablations for these design choices.

- <2> Robustness of semantic clustering. I am concerned about using K-means in an extremely high-dimensional space. If I understand the procedure correctly, only the $G$ rollouts are clustered (with $G=8$), yielding a very small sample; clustering eight points with Euclidean K-means is statistically fragile: initialization and small perturbations can change assignments and thus the entropy. The authors provide neither low-dimensional projections nor cluster-quality statistics, so it is unclear whether the reported entropy reflects genuine semantic modes or K-means noise on tiny samples. Please report stability (e.g., across seeds and across groups over training), cluster-quality metrics (e.g., silhouette, Calinski–Harabasz, Davies–Bouldin), and ablations over $G$.


- <3> The ablations on major modules are unclear. For several benchmarks (e.g., MATH500, AMC, Minerva, Olympiads; Fig. 3(a)/(b)), I cannot see how each module contributes to the final results. Some modules even decrease performance, which weakens the rationale behind the design. In addition, why did you use different base models across the two ablation sets? Please also include DAER/TDAR ablations on 7B models.

- <4> The baseline performance seems unusual. For example, how does Qwen-Math-7B achieve 74.0 pass@1 on MATH500? You are using pass@1 with greedy decoding (i.e., $\text{temp}=0$), right? The same baseline for Qwen-Math-7B should be around 65%, which aligns with a published COLM 2025 result (64.3% in https://arxiv.org/pdf/2504.07086). This discrepancy raises concerns about the evaluation protocol, and I worry that subsequent works might cite a potentially misreported result if it appears in a top-tier venue such as ICLR.

- <5> **Benchmark results are misleading and unfair**. For example, Dr. GRPO was trained on the MATH dataset (see p7 of https://arxiv.org/pdf/2503.20783), whereas your model was trained on the DAPO dataset, with different training steps and source prompts. It is not appropriate to copy Dr. GRPO’s results into your table and compare them without matching the training configuration. This is merely one part; the whole evaluation and benchmark protocols are highly concerned.

**Questions:**

See weakness.

---

### Official Review · Reviewer_3i9p · 2025-10-31

**Soundness:** 3
**Presentation:** 2
**Contribution:** 2
**Rating:** 2
**Confidence:** 4

**Summary:**

This paper proposes ZAPO, a reinforcement learning method that reuses “zero-advantage” samples in RLVR training instead of discarding them. By introducing adaptive entropy rewards and dynamic advantage reshaping, ZAPO turns these ineffective samples into useful signals, improving both training efficiency and reasoning performance. Experiments on multiple math reasoning benchmarks with Qwen-based models show consistent gains over GRPO and GRESO.

**Strengths:**

1. Offers a practical modification to RLVR by activating zero-advantage samples rather than discarding them outright.

2. Experimental results show improvements across several reasoning benchmarks.

**Weaknesses:**

1. Limited Novelty: The idea of using entropy-based rewards to encourage exploration is conceptually similar to prior entropy-based RL methods (e.g., [1-2]).

2. Incomplete Baseline Coverage: The paper does not compare ZAPO against other entropy-based or exploration-oriented RLVR methods.

3. Evaluation Gaps: Lacks pass@k or exploration-oriented metrics to substantiate claims about expanded “capability boundaries.”

4. Model Scope: Only Qwen-family models are tested; no validation on other architectures like LLaMA or DeepSeek-native models.

5. Benchmark Scope: Missing more recent reasoning benchmarks (eg, AIME25, HMMT-Feb25, CMIMC25).

6. Data contamination Risk: Results may be affected by potential contamination issues in Qwen2.5 evaluations [3].

**Questions:**

1. How does ZAPO perform compared to other entropy-based RLVR methods (e.g., [1,2])?

2. Does the proposed method improve pass@k (k>1) metrics, indicating real exploration gains?

3. Can the method generalize to other LLM families (e.g., Llama, DeepSeek)?

4. How does ZAPO perform on newer benchmarks (AIME25, HMMT-Feb25)?


[1] Cui et al. The entropy mechanism of reinforcement learning for reasoning language models. arXiv preprint arXiv:2505.22617

[2] Cheng at al.  Reasoning with exploration: An entropy perspective. arXiv preprint arXiv:2506.14758

[3] Wu et al., Reasoning or Memorization? Unreliable Results of Reinforcement Learning Due to Data Contamination, arXiv:2507.10532

---

### Official Review · Reviewer_wsPA · 2025-11-03

**Soundness:** 3
**Presentation:** 3
**Contribution:** 3
**Rating:** 6
**Confidence:** 3

**Summary:**

This paper introduces ZAPO (Zero-Advantage sample-augmented Policy Optimization), a method that leverages zero-advantage samples in RLVR to improve training efficiency and model reasoning. By applying dual-level adaptive entropy rewards (DAER) and temporal dynamic advantage reshaping (TDAR), ZAPO activates these samples, leading to better performance on math reasoning tasks and faster training compared to existing methods like GRPO and GRESO. Experimental results demonstrate significant improvements in both reasoning and efficiency across multiple datasets.

**Strengths:**

1. The proposal of leveraging zero-advantage samples rather than discarding them is a novel contribution to RLVR methods, which represents a novel and impactful contribution.

2. The paper includes a complete ablation study that effectively demonstrates the importance of all key components like DAER and TDAR in improving the model’s performance.

3. The improvements shown in Figure 1 and Table 1 demonstrate both the reasoning and efficiency gains offered by ZAPO compared to existing methods.

**Weaknesses:**

1. The paper does not analyze how entropy changes during training. Given that ZAPO relies on entropy-based rewards, it would be important to see if the entropy evolves as expected throughout training.

2. The paper sets the max training steps to 1000, but experiments only run for 160 or 320 steps. The necessity of setting max steps to 1000 steps is not well explained, and it raises the question of whether ZAPO and the other methods have all fully converged by this point and whether ZAPO possesses a higher performance upper bound than that of the other methods.

3.  Figure 2 shows that zero-advantage samples make up over 25% of the data. If these samples were fully utilized, the expected speedup compared to discarding them would be 1.33×. However, ZAPO only achieves a speedup of 1.2× and 1.3× in some settings, as shown in Table 2. How does this limited speedup demonstrate the effectiveness of ZAPO in utilizing zero-advantage data? Does this suggest that ZAPO is effectively leveraging these samples to improve reasoning, or is the primary benefit just faster training due to the addition of zero-advantage data?

**Questions:**

1. If the dual-level adaptive entropy reward mechanism were applied to more rollout data beyond zero-advantage samples, how would this affect the model’s performance? Would it enhance exploration and reasoning further, or could it lead to over-exploration or other unintended consequences?

---

### Official Review · Reviewer_KEkh · 2025-11-04

**Soundness:** 2
**Presentation:** 3
**Contribution:** 2
**Rating:** 4
**Confidence:** 3

**Summary:**

This paper proposes ZAPO, a modification to GRPO (Group Relative Policy Optimization) used in reinforcement learning with verifiable rewards (RLVR). GRPO often encounters zero-advantage groups—cases where all rollouts for a prompt receive identical rewards, giving zero gradients. ZAPO’s premise is that rather than discarding those groups, we can “activate” them so they still contribute to learning.

The method adds two components:

Dual-level Adaptive Entropy Reward (DAER): adds entropy-based pseudo-rewards for zero-advantage groups. Easy prompts (all correct) receive positive entropy rewards to encourage output diversity; hard prompts (all wrong) receive negative entropy shaping to guide more stable reasoning paths.

Temporal Dynamic Advantage Reshaping (TDAR): increases the gradient weight of harder prompts as training progresses.

The authors evaluate ZAPO on five math-reasoning datasets and three LLMs, reporting modest but consistent Pass@1 gains and reduced wall-clock training time due to fewer steps.


Disclosure: I used assistive writing tools to draft this review; all evaluations and judgments are my own.

**Strengths:**

Clear problem statement and simple mechanism. Converting “dead” groups into gradients is a straightforward way to reduce compute waste in GRPO-style training. DAER/TDAR integrate cleanly without a value model.

Consistent results across models/datasets. The paper shows average Pass@1 improvements on all three backbones and five math benchmarks, with especially strong numbers on AIME24.

Step/clock efficiency. Despite higher per-step cost (entropy + clustering), fewer steps are needed, yielding overall wall-clock reductions relative to GRESO.

Ablations are directionally informative. Removing DAER or TDAR hurts, and using both semantic and token-level entropy is better than either alone.

**Weaknesses:**

Problem magnitude is not quantified, and evidence is limited.
The only prevalence evidence is Figure 2, a single curve showing the ratio of zero-advantage groups for GRESO only. It spikes early (~0.4) and then stabilizes around ~0.26–0.30. There are no error bars, no numerical tables, no GRPO or ZAPO comparison, and no split of easy (all-1) vs hard (all-0.1). As presented, zero-advantage appears moderate rather than catastrophic; the paper does not establish that this inefficiency is pervasive enough to motivate a new algorithm. it's only 1 plot even if the authors believe 0.4/0.3 is significant.

Attribution is missing: hard-case activation vs easy-case diversity.
DAER intentionally mixes two different goals: (a) reward diversity on prompts the model already solves (easy), and (b) inject signal into all-wrong prompts (hard). The paper never separates their contributions. Without a hard-only DAER condition (turn off easy-case shaping) and a conversion metric (all-wrong → mixed/at-least-one-correct groups), it remains unclear whether ZAPO actually extends capability or mainly encourages stylistic variation on solved prompts. I think this is really important because the main advantage of RL is that it extends capabilities "beyond the data". But if it's just rephrasing then that would explain the low multipliers of improvement in the abstract. I think the most essential data is to show novel "beyond capacity" is what this algorithm leverages most. If that leads to low improvements that's surprising but it is **the best scenario** -- to inject true new information. Which there is no evidence. If this point is addressed, then I'd potentially increase the score.

Potential train/eval overlap on Math500.
The training recipe uses Light-Eval; Math500 is described as a random subset of Light-Eval. The paper should explicitly guarantee that training excludes the Math500 subset or provide a filtered evaluation to avoid contamination. Or at least it wasn't clear to me.

Statistics are thin.
Evaluations are repeated 4×, but the paper does not report confidence intervals or significance tests/p-values. This is particularly important on AIME24 (n=30), where variance is high.

Scope is narrow; stronger generalization tests are absent.
All tasks are math-reasoning. There is no evaluation on more stringent modern suites such as Putnam-AXIOM and its functional Variation set (built to probe functional invariance and contamination). Given the claim of expanding capability and reducing “entropy collapse,” performance on such unseen-variation splits would be highly probative. If you include results on this benchmark and it improves I'd be very tempted to increase the score.

I feel this is mostly useful only to frontier labs that have to do RL to make customers happy and burn gpus time, and thus very incremental. But still useful I suppose.

**Questions:**

Quantify prevalence and impact. Report the exact fraction of zero-advantage groups over training for GRPO, GRESO, and ZAPO, with error bars and an easy vs hard split; show how ZAPO changes these ratios.

Eliminate contamination. Clarify and, if needed, re-run to ensure no train/eval overlap between Light-Eval training and Math500 test.

Statistics. Provide CIs (e.g., paired bootstrap) and basic significance tests for all headline numbers; de-emphasize small-n sets unless aggregated over more seeds/splits. No need to train more, just use the benchmark eval size to estimate SE --> CI. And p-values.

Generalization stress test. Evaluate on Putnam-AXIOM (Original and Variation) and report both absolute scores and the Original→Variation drop. Demonstrating smaller drops or preserved gains would substantiate real capability improvements.

PS: minor typos to be fixed.

PS2: although I gave a 4, I was really tempted to give a 6. Doing some of the above to some quality would convince me to raise my score (only because I think leveraging zero advantage is a good idea and actually useful in practice, though, not super novel and arguably incremental). Depending on how well the remaining points are addressed if they are I might increase my score more.

---

### Official Review · Reviewer_vTGb · 2025-11-10

**Soundness:** 3
**Presentation:** 2
**Contribution:** 1
**Rating:** 2
**Confidence:** 4

**Summary:**

This paper tackles the inefficiency caused by zero-advantage samples in RLVR (Reinforcement Learning with Verifiable Reward) for LLMs, especially with the GRPO algorithm. These samples—from problems either fully mastered or beyond the model’s capacity—cause gradient vanishing and waste computation. As traditional methods only filter them, the authors propose **ZAPO**, which activates these samples via two key mechanisms DAER and TDAR.

**Strengths:**

1. Breaks from the traditional "discard zero-advantage samples" paradigm, instead proposing to *leverage* them—turning a computational burden into a valuable resource for training, which is a conceptual innovation in RLVR.
2. The paper is well-structured and addresses a real-world problem.

**Weaknesses:**

1. The indicators **SE** and **TE** in **DAER** do not exhibit a clear correlation with accuracy. The design principle that *higher entropy in positive samples yields greater rewards, while higher entropy in negative samples incurs heavier penalties* lacks theoretical grounding and systematic justification.These aspects are largely heuristic in nature and lack solid theoretical or empirical support. Moreover, the idea of *rewarding higher entropy in positive samples* has already appeared repeatedly in prior works.
2. **DAER** and **TDAR** are not conceptually connected; rather, they appear to be two loosely coupled components assembled together.
    In Figure 3, the *w/o TDAR* variant even outperforms **ZAPO** on several test datasets, which makes it difficult to substantiate the claimed effectiveness of the proposed method.
3. The performance improvement reported in the paper is limited. On both **Qwen2.5-Math-base-1.5B** and **Qwen2.5-Math-base-7B**, the proposed approach achieves less than a 1% average accuracy gain compared with the second-best model. Moreover, the paper does not provide any comparison of **time overhead** relative to other baselines, even though the proposed method requires an additional **embedding model** and **clustering process**. The experiments do not demonstrate the motivation of this work — for example, how **TE** and **SE** influence or determine the response quality, or whether the **TDAR** strategy is indeed effective. In the appendix experiments, only a comparison with **GRESO** is presented, leaving out evaluations against other representative baselines. Additionally, the proposed method introduces numerous hyperparameters, yet their sensitivity has not been thoroughly analyzed.

**Questions:**

1. It remains unclear which of **TE** or **SE** serves as a more reliable indicator for evaluating the quality of model responses.
2. Does the proposed method have any accelerating effect on the model’s training process?

---

### Note · Authors · 2025-12-01

I have read and agree with the venue's withdrawal policy on behalf of myself and my co-authors.